# *Staphylococcus aureus* Lung Infection Results in Down-Regulation of Surfactant Protein-A Mainly Caused by Pro-Inflammatory Macrophages

**DOI:** 10.3390/microorganisms8040577

**Published:** 2020-04-16

**Authors:** Elisabeth Schicke, Zoltán Cseresnyés, Knut Rennert, Vanessa Vau, Karoline Frieda Haupt, Franziska Hornung, Sandor Nietzsche, Fatina Swiczak, Michaela Schmidtke, Brigitte Glück, Mirijam Koch, Michael Schacke, Regine Heller, Alexander S. Mosig, Marc Thilo Figge, Christina Ehrhardt, Bettina Löffler, Stefanie Deinhardt-Emmer

**Affiliations:** 1Institute of Medical Microbiology, Jena University Hospital, Am Klinikum1, D-07747 Jena, Germany; elisabeth.schicke@uni-jena.de (E.S.); mirijam.koch@med.uni-jena.de (M.K.); bettina.loeffler@med.uni-jena.de (B.L.); 2Section of Experimental Virology, Institute of Medical Microbiology, Jena University Hospital, Hans-Knöll-Str. 2, D-07745 Jena, Germany; vanessa.vau@med.uni-jena.de (V.V.); Karoline.haupt@med.uni-jena.de (K.F.H.); franziska.hornung@uni-jena.de (F.H.); michaela.schmidtke@med.uni-jena.de (M.S.); glueck.brigitte@t-online.de (B.G.); michael.schacke@med.uni-jena.de (M.S.); christina.ehrhardt@med.uni-jena.de (C.E.); 3Applied Systems Biology, Leibniz Institute for Natural Product Research and Infection Biology—Hans Knöll Institute, Adolf-Reichwein-Straße 23, 07745 Jena, Germany; zoltan.cseresnyes@hki-jen.de (Z.C.); Thilo.Figge@hki-jena.de (M.T.F.); 4Dynamic42, GmbH, Winzerlaer Str. 2, D-07745 Jena, Germany; knut.rennert@dynamic42.com; 5Center for Electron Microscopy, Jena University Hospital, D-07743 Jena, Germany; sandor.nietzsche@med.uni-jena.de; 6Institute of Biochemistry, Jena University Hospital, D-07743 Jena, Germany; Fatina.swiczak@med.uni-jena.de (F.S.); Alexander.Mosig@med.uni-jena.de (A.S.M.); 7Faculty of Biological Sciences, Friedrich Schiller University of Jena, D-07745 Jena, Germany; REGINE.HELLER@med.uni-jena.de; 8Institute of Microbiology, Faculty of Biological Sciences, Friedrich Schiller University Jena, 07743 Jena, Germany

**Keywords:** *Staphylococcus aureus*, pneumonia, surfactant protein-A, influenza A virus, human alveolus-on-a-chip

## Abstract

Pneumonia is the leading cause of hospitalization worldwide. Besides viruses, bacterial co-infections dramatically exacerbate infection. In general, surfactant protein-A (SP-A) represents a first line of immune defense. In this study, we analyzed whether influenza A virus (IAV) and/or *Staphylococcus aureus* (*S. aureus*) infections affect SP-A expression. To closely reflect the situation in the lung, we used a human alveolus-on-a-chip model and a murine pneumonia model. Our results show that *S. aureus* can reduce extracellular levels of SP-A, most likely attributed to bacterial proteases. Mono-epithelial cell culture experiments reveal that the expression of SP-A is not directly affected by IAV or *S. aureus*. Yet, the mRNA expression of SP-A is strongly down-regulated by TNF-α, which is highly produced by professional phagocytes in response to bacterial infection. By using the human alveolus-on-a-chip model, we show that the down-regulation of SP-A is strongly dependent on macrophages. In a murine model of pneumonia, we can confirm that *S. aureus* decreases SP-A levels in vivo. These findings indicate that (I) complex interactions of epithelial and immune cells induce down-regulation of SP-A expression and (II) bacterial mono- and super-infections reduce SP-A expression in the lung, which might contribute to a severe outcome of bacterial pneumonia.

## 1. Introduction

Pneumonia is the most severe inflammatory disease of the lower respiratory tract and also the most common infectious disease worldwide. A wide variety of microorganisms can induce pneumonia, including viruses and bacteria. Among these, influenza A and B viruses (IAV, IBV) are primary causative agents [1]. The high mortality rates during the large influenza virus (IV) epidemics and pandemic outbreaks are associated with bacterial super-infections, and here *Staphylococcus aureus (S. aureus)*, a facultative-pathogenic bacterium, is of major importance. *S. aureus* colonizes epithelial surfaces, but can also cause a broad spectrum of infections ranging from superficial skin infections to life-threatening diseases, such as bacterial pneumonia [2].

The first line of defense during infections, apart from physical barriers, is regulated by an innate host immune response in which white blood cells and chemical components play a crucial role [3]. Granulocytes and phagocytes are the key cells of the innate immune system [4]. Besides, the non-specific cellular response, including mucous and also surfactant proteins, are principal targets to fight pathogens [5]. Pulmonary surfactant has long been known to be essential to lower the surface tension at the air-liquid interface of the lung and to prevent its collapse at end-expiration. However, surfactant proteins have also immunomodulatory and antimicrobial activity and thereby are important agents in primary host immune defense [6,7,8].

Surfactant proteins interact with several pathogens, among others, with IAV [9] and *S. aureus* [10]. During the IAV invasion into the lung, the viral hemagglutinin binds to sialic acid residues on the surface of epithelial cells to initiate viral internalization [11]. Surfactant protein-A (SP-A) also presents sialic acid residues to bind IAV [12]. The agglutination of SP-A reduces the infectivity and dissemination of the virus and supports its clearance by immune cells [9]. Furthermore, SP-A promotes aggregation and phagocytosis of *S. aureus* by neutrophils and macrophages and inhibits bacterial adherence to the epithelial layer, thus preventing the bacterial penetration into the host cells [13]. 

Several inflammatory cytokines alter SP-A expression. Among these, TNF-α is known as a strong suppressor of SP-A [14,15,16,17]. Immune cells especially, such as macrophages, are known to release large amounts of TNF-α during bacterial infections [18,19,20,21]. 

Yet, bacteria can also directly affect pulmonary surfactant proteins [22], as *S. aureus* expresses a cysteine protease staphopain A (ScpA, sspA) that cleaves SP-A and inactivates the antibacterial activity of the protein, promoting the colonization of the lung by *S. aureus* and other pathogens [23]. 

Taken together, many parameters during an infection affect the SP-A expression. In this study, we aimed to elucidate whether IAV, *S. aureus* or the combined infection of both pathogens has an impact on the SP-A expression in the lung, which could represent a mechanism to weaken the host defense system and worsen the outcome of a lung infection. 

## 2. Materials and Methods

### 2.1. Cell Culture

For the single cell-culture, we use alveolar epithelial cells. The human adenocarcinoma cell line NCI-H441 (ATCC, Manassas, VA, USA) was cultivated in RPMI-medium (Roswell Park Memorial Institute medium, Gibco, Thermo Fisher Scientific, Waltham, MA, USA) and supplemented with 10% inactivated fetal bovine serum (FBS, Sigma Life Science, Merck, Darmstadt, Germany). The cell line was sub-cultured in the laboratory and tested for contamination with mycoplasma. NCI-H441 cells were grown up to 80% confluence before being sub-cultured and used until the 18th passage.

Besides this, primary monocytes were isolated from whole peripheral blood of healthy donors. The blood was diluted in an equal volume with phosphate buffer saline (PBS, 2 mM EDTA, 0.1% FBS (Sigma Life Science, Merck, Darmstadt, Germany) and then gently transferred to the surface of Biocoll separation solution (Biochrome GmbH, Berlin, Germany). The monocytes were separated from blood cells by centrifugation without bakes. Cells were washed with PBS three times with centrifugation steps in-between. For cell culture experiments, 6 × 10^6^ monocytes/well were cultured in 24-well dishes in RPMI-medium supplemented with 10% autologous human serum, 10 ng/mL human granulocyte-macrophage colony-stimulation factor (GM-CSF, PeproTech, Hamburg, Germany), 10 ng/mL macrophage colony-stimulation factor (M-CSF, PeproTech, Hamburg, Germany), and penicillin/streptomycin. The cells were incubated for 1 h at 37 °C and 5% CO^2^ and then washed twice with RPMI. Cells were cultured for 6 days. A medium exchange was performed every second day. RPMI-medium was supplemented with 10% autologous human serum, 10 ng/mL GM-CSF, to induce macrophage differentiation, and penicillin/streptomycin.

Macrophage differentiation from monocytes was routinely checked by staining for macrophage cell type markers CD68, and CD163 in random-wise selected control samples. No significant donor-related variation of differentiation efficiency or cell numbers was observed. 

### 2.2. Human Alveolus-on-a-Chip Model

For the establishment of the human alveolus-on-a-chip model, we used MOTiF biochips, manufactured and purchased by microfluid ChipShop GmbH (Jena, Germany). The chip consists of two chambers that are separated by an 11 µm thick polyethylene-terephthalate (PET) membrane with a pore diameter of 8 µm and pore density pf 2 × 10^5^ pores/cm^2^. The capacity of the endothelial is 220 µL (including afferent and efferent channels), and the volume of the epithelial chamber is 120 µL (also including the channels). Peristaltic pumps were used for the perfusion of the biochip, as explained previously [24].

For the cell cultivation, we use NCI-H441 cells, endothelial cells (human umbilical vein endothelial cells, HUVECs) and human monocyte-derived macrophages (hMdM) as described above cm^2^ [25].

NCI-H441 cells (ATCC, Manassas, VA, USA) were seeded on the opposite side of the membrane (epithelial chamber) with a density of 2.7 × 10^5^ NCI-H441/cm^2^. Cells were cultured for 3 d until confluency with a daily media exchange. The NCI-H441 medium was supplemented with 1 µM dexamethasone (Sigma-Aldrich, Munich, Germany). 

After 1 week of culture, hMdM cells were seeded on top of the NCI-H441 cell layer with a density of 0.9 × 10^5^ macrophages/cm^2^ (Appendix A). The random-wise analysis of the macrophage density observed a typical monocyte differentiation rate of 90% without donor-specific variations. These data were obtained during the establishment of the human alveolus-on-a-chip model and were not determined for each chip. 

HUVECs were kindly provided by Regine Heller (Faculty of Biological Sciences, Friedrich Schiller University of Jena) and were isolated from human umbilical cord veins as described previously and seeded into the biochip (vascular cavity) at a density of 2.7 × 10^5^ HUVECs/cm^2^ in endothelial cell growth medium (PromoCell, Heidelberg, Germany) [26] for 48 h with a daily medium exchange [25]. 

### 2.3. Pathogens

For the propagation of IV, we used Madin-Darby canine kidney (MDCK) cells. MDCKs were cultured in EMEM (Eagle’s Minimum Essential Medium, ATCC, Wesel, Germany), supplemented with 10% fetal calve serum (FCS). For the virus propagation, the IV strain A/Puerto Rico/8 was passaged on MDCK in EMEM supplemented with 25 mM MgCl, 1.3% bicarbonate, and 0.333% trypsin. The infected cells were incubated at 37 °C and 5% CO_2_. After freezing and a centrifugation step, the virus was stored at −80 °C.

For the murine model, the A/H1N1/pdm09 influenza virus variant HA-D222-mpJena/5258, expressing an aspartate in position 222 of the viral hemagglutinin, was passaged on MDCK. HA-D222-mpJena/5258 was obtained after three plaque-purification steps [27] from the lung of BALB/c mice infected with the A/H1N1/pdm09 influenza virus isolate A/Jena/5258/09 (kindly provided by Andi Krumbholz [28].

The bacterial infection was performed with a methicillin-resistant *S. aureus*/USA300/wild type (WT), kindly provided by Lorena Tuchscherr [29]. The bacterial strain *S. aureus* USA300 was grown overnight in brain heart infusion (BHI)-medium at 37 °C with shaking. All bacteria were stored in the BHI medium at −80 °C. Colony-forming units (CFU) of *S. aureus* were determined weekly to calculate the number of infectious particles in the cryopreserved bacterial working solution and to exclude changes in the number of living bacteria by freezing.

The determination of the *S. aureus* load after the infection was performed by using the supernatants. For this, the supernatants were serial-diluted in PBS and plated on blood agar plates (Columbia blood agar, Oxoid, Germany). After overnight incubation at 37 °C, colonies were counted to detect the number of bacteria.

### 2.4. In Vitro Infection

For the viral infection of the NCI-H441 mono-cell culture as well as the infection on the human alveolus-on-a-chip model, cells were washed with PBS once ( Appendix A). The IV strain A/Puerto Rico/8 was added to the cells in RPMI (0.2% autologous human serum, 1 mM MgCl_2_, 0.9 mM CaCl_2_) and incubated for 30 min at 37 °C without centrifugation. For the specification of the pathogen concentration, we have used the multiplicity of infection (MOI). This information describes the ratio of pathogens to infected cells. For the viral infection of the NCI-H441 cells we use MOI 1. 

The bacterial infection on NCI-H441 cells was performed with RPMI (Roswell Park Memorial Institute 1640, Gibco, Thermo Fisher Scientific, Waltham, MA, USA) with 1% human serum albumin (HSA) and 1mM 4-(2-hydroxyethyl)-1-piperazineethanesulfonic acid (HEPES). The epithelial side of the human alveolus-on-a-chip model was infected with RPMI/BA (10% autologous human serum, 1 mM HEPES, 10 ng/mL GM-CSF) with bacteria (MOI 5). After the bacterial infection was performed, cells were incubated with bacteria for 90 min at 37 °C and 5% CO_2_. Afterwards, the cells were washed. To eliminate the non-internalized bacteria, cells were treated for 20 min with RPMI-medium containing lysostaphin at a concentration of 6 µg/mL for the NCI-H441 mono-cell culture and 20 µg/mL for the human alveolus-on-a-chip model. Subsequently, cells were washed with PBS and then incubated with RPMI for indicated time points at 37 °C with 5% CO_2_.

For the co-infection scenarios, both pathogens were added to the cells. For this, first, the infection with IV strain A/Puerto Rico/8 (MOI 1) was performed, followed by the bacterial infection with *S. aureus*/USA300/WT (MOI 5).

For the infection of the single cell-culture of hMdM, cells were isolated as described above and used after 6 days of cultivation. After a washing step with RPMI, the IV strain A/Puerto Rico/8 (MOI 1) was added to the cells in RPMI (0.2% autologous human serum, 1 mM MgCl_2_, 0.9 mM CaCl_2_) and incubated for 30 min at 37 °C without centrifugation until the indicated time points. For the single bacterial infection as well as the co-infection, we use *S. aureus*/USA300/WT (MOI 1).

The experimental setup always contained a cell control (designated as mock), a single infection with the virus, a single infection with bacteria, and a co-infection. All cells were treated with the same medium. This medium contained the according pathogen for the single- or co-infection or no pathogens for the mock-treated cells of the cell control.

### 2.5. In Vivo Infection

For the murine pneumonia model the mice were purchased from Janvier Labs (Le Genest-Saint-Isle, France). The animal application was proven by the local ethics committee of the Thuringian State Office for Consumer Protection (Thüringer Landesamt für Verbraucherschutz, Reg.-Nr.: UKJ-018-028, 10/10/2018). The mice were maintained according to institutional guidelines in individually ventilated cages, and food and water were given ad libitum. For the in vivo model, we use specific-pathogen-free (SPF) C57Bl/6 mice from Janvier Labs (Le Genest-Saint-Isle, France), providing us with an excellent health status for the animals. The animal facility of the Jena University Hospital also provides an excellent unit for performing animal experiments. Under S2 conditions, the animals are housed in separate units, which are monitored frequently.

Eight-week-old female C57bl/6 mice were infected with the IV strain HA-D222-mpJena/5258, *S. aureus* USA300 or with both pathogens in a co-infection. Therefore, the mice were anesthetized by isoflurane inhalation and intranasally inoculated with HA-D222-mpJena/5258 (5 × 10^4^ plaque-forming units (PFU)) in 20 μL NaCl or only 20 µl NaCl (mock-treated). The same method was used for *S. aureus* infection (1 × 10^7^ CFUs) two days later. During the infection period, the state of the animal health was monitored up to two times per day by controlling the weight, temperature and the scoring of the general behavior condition. Four days after HA-D222-mpJena/5258-infection, animals were sacrificed, and samples of blood and lung were subjected to further analysis. The viral titer and the bacterial counts of infected mice were determined in the right inferior lung lobe. 

Moreover, the right superior lung lobe was used to determine the mRNA expression of TNF-α and SP-A by quantitative real-time PCR. For histopathology, the left lung lobe of each mouse was fixed in 10% formalin for at least 24 h, dehydrated in a graded series of alcohol and xylene, and mounted in paraffin. For each lung sample, about ten 5 µm thick sections were stained with hematoxylin and eosin (HE) to study the degree of cellular infiltration. Sections of each lung were examined microscopically (Zeiss microscope Axio Vert.A1), and representative photographs were obtained (Axio Cam ERc5S).

### 2.6. Quantitative Real-Time PCR

For quantitative real-time PCR, cell infection was conducted as described above. Isolation of total RNA from macrophages, NCI-H441 and homogenized lungs was done by using the RNeasy Mini Kit (Qiagen, Hilden, Germany) according to the manufacturer’s instructions. Cells of the biochip model were first homogenized using the QIAshredder homogenizers (Qiagen) and total RNA was isolated with RNeasy Micro Kit (Qiagen) following manufacturer’s instructions. Equal amounts of RNA were transcribed into cDNA using QuantiNova Reverse Transcription Kit (Qiagen) according to the manufacturer’s protocol. Gene expression of several genes was determined by qRT-PCR on a Thermo Scientific™ PikoReal™ Real-Time PCR System using QuantiTect SYBR Green (Qiagen) and gene-specific primers (Table 1). Relative gene expression levels were referred to the housekeeping gene GAPDH for NCI-H441 and RPL37A for macrophages and calculated with the 2^−ΔΔCt^ method [30].

To analyze the mRNA expression of SP-A, we have compared the data from the infection with the pathogens and the data of the stimulation with recombinant TNF-α. For this, we used 10 nM of TNF-α (Gibco, Thermo Fisher Scientific, Waltham, MA, USA) added to RPMI. After 30 min of incubation on NCI-H441 cells, they were washed with PBS and incubated for 4 h. Afterward, the mRNA expression was performed as described above.

### 2.7. Protein Analysis 

For the detection of extracellular TNF-α we performed the infection of the cells with virus, bacteria and both pathogens as described above (compare with the chapter on in vitro infection) and determined the concentration of TNF-α within the supernatants.

Supernatants were collected at 30 min and 4 h p.i. and transferred to an antibody-coated 96-well plate in duplicates (Human TNF-α ELISA Kit, Invitrogen, Thermo Fisher Scientific, Waltham, MA, USA) and subsequently analyzed according to the manufacturer’s instructions. For the identification of SP-A, we used NCI-H441-cells and performed the infection as previously described [25]. Next, 4 h and 18 h p.i. cells were lysed with radioimmunoprecipitation assay lysis buffer (RIPA (20 mM Tris-HCL (pH 7.4), 137 mM NaCl, 10% glycerol, 1% Triton X 100, 2 mM EDTA, 50 mM Naβ-Glycerophosphate, 20mM Sodium-Pyrophosphate, 0.2 mM Pefablock, 5 µg/mL leupeptin, 5 µg/mL aprotinin, 1mM sodium vanadate and 5mM benzamidine) for 30min at 4 °C. Centrifugation at 4 °C was utilized for clearing the cell lysate from debris. Afterwards, the protein concentration was determined using the Bradford method. For this, the concentration was adjusted to the lowest concentration with RIPA buffer and treated with Laemmli-buffer (Bio-Rad Laboratories, Inc. Hercules, CA, USA). The samples obtained this way were transferred in duplicates to a coated 96-well plate (Human SFTPA1/Surfactant Protein A ELISA Kit, Sandwich ELISA, LifeSpan BioSciences, Inc., Seattle, WA, USA) and the procedure was performed following manufacturer’s instructions. The optical density (OD)_450_ was determined by the microplate reader SPECTROstar Omega (BMG LABTECH, Ortenberg, Deutschland). The protein concentration was calculated by comparison to a standard curve. 

For the measurement of pro-inflammatory cytokines, a LEGENDplex™ Human Inflammation Panel (13-plex) (BioLegend, San Diego, CA, USA) was used. Alveolar biochips were infected as previously described and 4 h p.i. supernatants were collected. 25 µL of each sample was transferred in duplicates into the 96-well filter plate and the LEGENDplex panel was performed following the manufacturer’s instructions. Samples were measured the same day on a flow cytometer (BD, Accuri, BD Biosciences, Franklin Lakes, NJ, USA), and the cytokine amount was calculated by comparison to a standard curve. 

### 2.8. Immunofluorescence Microscopy

For the immunofluorescence staining of the cells of the human alveolus-on-a-chip model, the membranes with the NCI-H441 cells were fixed with 4% paraformaldehyde at room temperature. For the staining of the endothelial sides, cells were fixed for 15 min with ice-cold methanol at −20 °C. Afterwards, intracellular staining of SP-A was performed. For this, membranes were incubated overnight at 4 °C in PBS containing 0.1% Saponin, 3% donkey serum and anti-SP-A antibodies (Santa Cruz Biotechnologies, Dallas, TX, USA). Thereafter, fluorescent labeled secondary antibodies (donkey-anti-goat- AlexaFlour647, life technologies, Carlsbad, CA, USA) were used and incubated for 30 min. The stained samples were embedded into a mounting medium (S3023, Dako, Hamburg, Germany). Imaging was performed with an Axio Observer Z1 fluorescence microscope with ApoTome.2 extension (Carl Zeiss AG, Jena, Germany), and images were analyzed with ImageJ2 software (Fiji distribution).

### 2.9. Scanning Electron Microscopy

The fixation of the cells was performed inside the human alveolus-on-a-chip model by using a fixative solution (2.5% glutaraldehyde in cacodylate buffer) for 90 min. Afterwards, a fresh cacodylate buffer was used for 30 min to wash out the fixative. The membranes were dehydrated in ascending ethanol concentrations (30%, 50%, 70%, 90%, and 100%) for 15 min each. The samples were critical point dried using liquid CO_2_ and sputter coated with gold (thickness approx. 2 nm) using an SCD005 sputter coater (BAL-TEC, Balzers, Liechtenstein) to avoid surface charging. Finally, the specimens were investigated with a field emission (FE) SEM LEO-1530 Gemini (Carl Zeiss NTS GmbH, Oberkochen, Germany).

### 2.10. Image Analysis and Quantification

For the analysis, images were used as Z stacks in the Zeiss native image format “CZI”. After deconvolution with Huygens Software (Scientific Volume Imaging b.v.Hilversum, The Netherlands), the images were segmented in Imaris 9.2.1 (Bitplane, Zürich, Switzerland). The complete process was published before [25].

### 2.11. Statistical Analysis

All experiments were performed in technical duplicate and at least three were independent experiments. After an examination of quality criteria, non-parametrical methods were used for analysis. Statistical significances were evaluated by one-way-ANOVA (Kruskal–Wallis test) followed by Mann–Whitney U test or by using one-way-ANOVA Tukey’s multiple comparison test. Statistical analysis was performed using SPSS Statistics 25 software (IBM, Armonk, NY, USA) and Prism software (v.8; GraphPad Software, La Jolla, CA, USA).

## 3. Results

### 3.1. Neither IAV Nor S. aureus Directly Affects SP-A Expression in Epithelial Cells, But TNF-α is a Strong Inhibitor of SP-A Expression 

To determine the effect of the mono- and co-infections on the SP-A production we used single cell cultures of alveolar epithelial cells type II (NCI-H441 cells). To verify an efficient infection, viral titers (Figure 1A), as well as viable bacteria, were measured in a time-dependent manner (Figure 1B).

After mono-infection with IAV or *S. aureus* and after co-infection, SP-A mRNA-expression was not changed. By contrast, treatment with TNF-α, a well-known suppressor of SP-A, resulted in significantly decreased SP-A mRNA expression (Figure 1C). Next, we evaluated the protein levels of SP-A in epithelial supernatants. Here we were able to show that SP-A is significantly decreased after 18 h of bacterial mono- or co-infection (Figure 1D). This finding can be attributed to the cysteine protease staphopain A (ScpA, ssPA) that cleaves SP-A. We have verified the presence of ssPA in the bacterial strain *S. aureus* USA300 by using the PanStaph Alere Genotyping Kit (Figure 2). Here, we were able to demonstrate that *S. aureus* USA300 disposes of a large number of virulence factors. For instance, *S. aureus* strain USA300 expresses leucocidins (lukS, lukF), phenol-soluble modulin (PSM) [22], and superantigens (ssl) that are mainly directed against immune cells and induce strong proinflammatory and cytotoxic effects [31].

During viral and/or bacterial infection, the cytokine profile, including the expression of TNF-α, of infected cells is altered depending on the host cell type and on the pathogens. In an epithelial mono-cell culture (NCI-H441 cells) bacterial infection caused a much lower increase in TNF-α mRNA-expression compared to the infection of immune cells (hMdM) (Figure 3). Here, we measured up to a 6-log increase of the TNF-α mRNA-expression after bacterial infections. 

Macrophage differentiation from monocytes was proven by performing single-cell experiments with hMdM. Here we can demonstrate that macrophages react on their mRNA expression level after challenging with pathogens. All cells infected with virus, bacteria or both, had a significantly higher level of CD80 (mRNA) than the mock-treated cells (Appendix A). By contrast, a significant reduction of the M2 marker CD206 was observed after co-infection, indicating that this scenario represents the strongest challenge for the immune cells.

In summary, our results show that TNF-α expression after *S. aureus* infection was much higher in macrophages than in epithelial cells, whereas IAV-infection did not significantly increase the TNF-α expression in either cell types (Figure 3). 

### 3.2. Interaction of Epithelial, Endothelial and Macrophages Causes SP-A Downregulation Mainly Triggered by Bacteria

Within the human alveolus-on-a-chip model, epithelial cells (NCI-H441), endothelial cells (HUVECs) and macrophages (hMdM) were co-cultured under perfusion of the endothelial side and infected via the epithelial side with the pathogens as indicated. The successful viral infection was proven by the budding process of IAV, as visible in the SEM-pictures (Figure 4A). Furthermore, we could visualize the adherence of *S. aureus* on the epithelial surface within the alveolus-on-a-chip model (Figure 4A). In this model system, composed of the different cell types, we measured elevated levels of TNF-α caused by *S. aureus* mono- or IAV-coinfection, whereas IAV did not enhance TNF-α release (Figure 4B). 

To evaluate the impact of the infection and subsequent TNF-α expression on the production of SP-A, we measured the SP-A protein content by immunofluorescence staining in the human alveolus-on-a-chip model following infection with IAV, *S. aureus* or co-infection (Figure 4C). By using quantitative analysis of the mean fluorescence intensity (MFI) we detected decreased levels of SP-A following infection by *S. aureus* and in the co-infection scenario (Figure 4D). 

### 3.3. S. aureus Downregulates SP-A Expression in a Murine Model of Pneumonia

Finally, to reproduce the results obtained in the cell-culture systems in an in vivo murine lung infection model, we established a pneumonia model and infected the mice with IAV, *S. aureus* or with both pathogens. After day 4 mice were sacrificed and the bacterial counts in the lung tissue were determined. Furthermore, the mRNA-expression of SP-A was quantified. 

Our results show a high viral load (PFU/mL) 4 days p.i. (Figure 5A). Similarly, bacterial infection resulted in detectable CFUs within the lung tissue of the mouse (Figure 5B). Comparable to our in vitro results, we could clearly show that the SP-A expression within the bacterial mono- and co-infection was down-regulated four days after infection, whereas IAV did not cause a down-regulation in SP-A expression (Figure 5C). To further analyze the effect of viral and/or bacterial infection in the murine model system, we performed histopathological staining of the lung tissue. In mock-treated mice we recognized a fine and healthy alveolar structure. In infected animals, we detected an inflammatory infiltration of the alveolar structures that was caused by a bacterial, viral or co-infection (Figure 5D). These results indicate that infections can trigger lung inflammation by different mechanisms that can be dependent (*S. aureus* infection) or independent (IAV infection) on SP-A expression.

## 4. Discussion

Secondary bacterial pneumonia is one of the main reasons for increased morbidity and death rates after IAV infections. Apart from the pathogen load caused by the super-infection, the dysregulated immune response is also involved in the severe outcome of concomitant bacterial and viral infections. The pulmonary surfactant plays an important role as the first line of defense. The aim of our study was to elucidate whether a viral/bacterial co-infection has an impact on SP-A expression, the major surfactant protein.

SP-A fulfils many functions in the innate immune host defense. It regulates the inflammatory chemotaxis via the collagen-like region which induces an increased macrophage migration. Furthermore, SP-A is able to enhance phagocytosis as well as killing by macrophages and neutrophils via agglutination and opsonization of pathogens [14].

The deficiency of surfactant proteins can lead to various diseases. First of all, in premature infants, the insufficient surfactant production causes neonatal respiratory distress syndrome (RDS) [32]. A decreased SP-A level in amniotic fluid of the mother seems to be associated with an increased risk for the infant to be born with RDS [33]. Furthermore, infants with RDS have decreased SP-A level in tracheal aspirates [34]. There are two therapeutic approaches in imminent premature birth. The prenatal administration of corticosteroids, particularly Betamethasone, accelerates lung maturation [35] and increases the amount of surfactant [36,37] and especially SP-A and SP-B [38] in the fetal lung. The second therapy strategy is the postnatal administration of surfactant [39]. Early administration of surfactant reduces the risk of pneumothorax and decreases mortality rates in preterm children [40]. Particularly the combined use of corticosteroids and surfactant therapy improves the outcome of infants with RDS [41].

Furthermore, the adult acute respiratory distress syndrome (ARDS) is also associated with a decreased amount of SP-A and surfactant phospholipids [42]. In agreement with this result patients with respiratory failure aggravated by sepsis also had a decreased surfactant level [43] indicating that infections can worsen the already low surfactant amount in patients with ARDS. Additionally, decreased SP-A levels were found in the bronchoalveolar lavage (BAL) of patients with pneumonia and without ARDS [44,45]. 

These examples demonstrate the great importance of a sufficient surfactant level in the lung and also the possibility to improve the lung function by external surfactant administration during a state of deficiency.

*S. aureus* disposes of extracellular proteases with a wide variety of virulence-associated functions. Here, metallo-, cysteine and serine protease lead to tissue degradation and interference of the host immune response. The protease ssPA was described as the most efficient in degrading SP-A, even in a nanomolar concentration [23]. The direct effect of extracellular *S. aureus* proteases impairs the bacterial clearance and results in enhanced adherence and invasion of *S. aureus* to epithelial cells [21]. Additionally, surfactant proteins affect IAV infection [12]. However, an interfering effect of IAV infection on SP-A has not been described.

The regulation of SP-A is dependent on cytokines, in particular on TNF-α. TNF-α is capable of altering the mRNA level of SP-A via the phosphorylation of p38 mitogen-activated protein kinases [46,47]. As a consequence, an increased TNF-α level during an *S. aureus* infection hampers the function of SP-A. It is well known that *S. aureus* infections lead to a high release of TNF-α, in particular originating from immune cells, such as macrophages. Yet, TNF-α expression was not increased upon IAV infection in immune cells within the present experimental setting. 

Although SP-A is mainly produced by alveolar-epithelial type II cells, other cell types, such as endothelial and immune cells, are in close contact with each other within the alveolus. The different cell types interact via cytokines and influence each other, which cannot be reproduced in vitro in a mono-cell culture system. Consequently, the response to infection in the lung is highly complex and requires appropriate model systems where different cell types interact with each other. Frequently used models to study infections are mice. Here, the infection can be analyzed in the complete organ, including damages in the alveolar structures. Yet, the mice models have some disadvantages. (i) The cells are of murine origin and in particular immune functions can largely differ from human cells. For example, the action of defined bacterial toxins (e.g., superantigens and leucocidins) can be very species-specific and largely restricted to human cells [48,49,50]. (ii) In the murine model it is difficult to attribute the host response to a defined cell type. To counter these disadvantages, we recently published a human-alveolus-on-a-chip model that is composed of defined cell types (epithelial, endothelial and immune cells) of human origin [25]. 

By combining the different model systems in our present study, we demonstrate that *S. aureus* reduces SP-A protein levels, as *S. aureus* expresses the cysteine protease sspA which is able to cleave SP-A [21]. However, neither IAV nor *S. aureus* directly affects the mRNA expression of SP-A in NCI-H441 mono-cell culture, indicating that bacterial and viral pathogens do not affect the SP-A expression within the epithelial cells. Furthermore, our results demonstrate that TNF-α is a strong inhibitor of SP-A mRNA-expression. Bacterial mono- or co-infection strongly induce the expression of TNF-α, in particular in hMdM cells. By contrast, IAV infection has only a minor effect on TNF-α synthesis, which can be explained by the fact that IAV is inhibited by TNF-α and therefore TNF-α is disadvantageous for the survival of the virus particles [51]. The interaction of epithelial, endothelial and immune cells within the human alveolus-on-a-chip model demonstrates high amounts of TNF-α in response to bacterial infection. Finally, through MFI quantification we could demonstrate that bacterial infection significantly decreased intracellular SP-A protein levels. 

Additionally, we have performed in vivo studies. Mice were infected with IAV, *S. aureus* or both pathogens in the lung that caused inflammation in all cases. In line with our in vitro results, we could clearly demonstrate that the SP-A mRNA expression was down-regulated, particularly in the lung tissue of the bacterial-infected mice. 

Our results show that beside the direct effect of *S. aureus* leading to the cleavage of extracellular SP-A, there is an indirect effect based on infection via the cytokine production, notably by TNF-α. During infection, macrophages are one of the main sources for elevated TNF-α levels that have already been described to lower mRNA levels of SP-A. 

Therefore, we concluded that an interaction of macrophages and alveolar epithelial cells decrease SP-A mRNA levels. These data demonstrate that the analysis of infection processes requires the use of sophisticated model systems that are based on various human cells and mimic the organ system.

## 5. Conclusions

Pneumonia is the leading cause of morbidity and mortality worldwide and viral/bacterial co-infection worsens the severity of the disease. The innate immune response is a crucial part of the defense against these pathogens. Surfactant proteins are a part of this mechanism and have a major effect on the alveolar homeostasis.

Our study investigated the impact of IV and/or *S. aureus* infection on the SP-A expression in vitro and in vivo. We can show that complex interactions of epithelial cells and immune cells result in the down-regulation of SP-A mRNA expression. Furthermore, bacterial mono- and super-infections reduce the SP-A expression in a murine model which might contribute to the severe outcome of bacterial pneumonia.

## Figures and Tables

**Figure 1 microorganisms-08-00577-f001:**
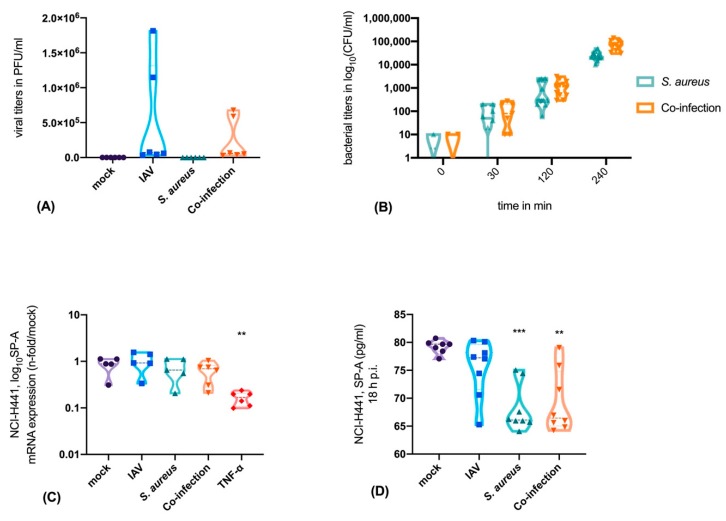
NCI-H441 cells were productively infected with influenza A virus (IAV) upon mono- and co-infection with *S. aureus* USA300 (**A**). The bacterial load increases over time upon mono- and viral co-infection (**B**). The infection of epithelial cells does not affect the mRNA expression of SP-A, whereas TNF-α reduces the SP-A expression (**C**). Extracellular SP-A levels are decreased in the supernatants of the epithelial cell culture system after *S. aureus* infection, as well as co-infection of *S. aureus* and IAV (**D**). Mock-treated cells obtained the same medium and treatment as infected cells without the pathogen. Ordinary one-way ANOVA, Tukey´s multiple comparison, ** *p* < 0.01, *** *p* < 0.001.

**Figure 2 microorganisms-08-00577-f002:**
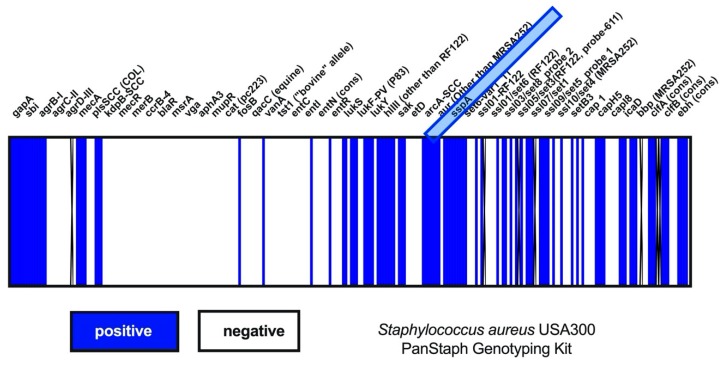
Three hundred and thirty genes of *S. aureus* USA300 were monitored by use of PanStaph Genotyping Kit (Alere, Germany). Blue graphs show the detectable genes as positive and the white bars as negative. The virulence factor ssPA was positively detectable within the *S. aureus* strains USA300. Every fifth gene was named within the heat map.

**Figure 3 microorganisms-08-00577-f003:**
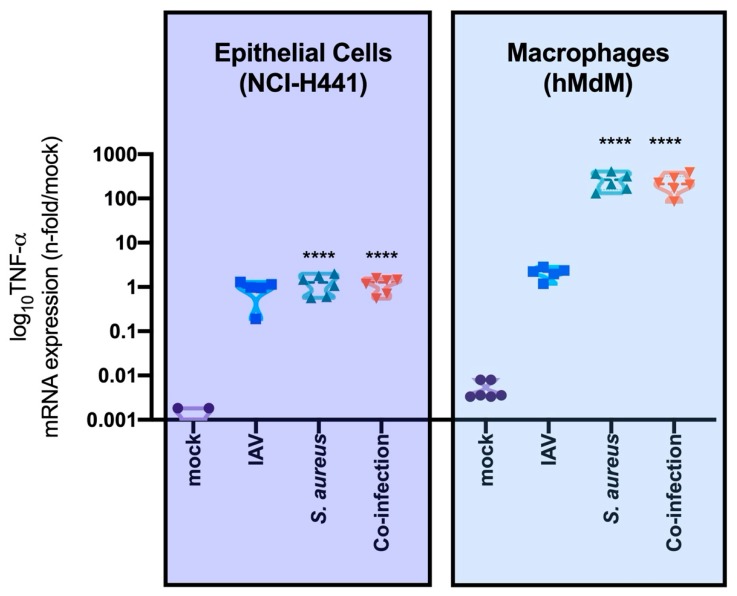
TNF-α expression is significantly higher in immune cells (hMdM) than in epithelial cells (NCI-H441) after bacterial mono- or co-infection. Mock-treated cells obtained the same medium and treatment as infected cells without the pathogen. Two-way ANOVA, multiple comparisons, **** *p* < 0.0001.

**Figure 4 microorganisms-08-00577-f004:**
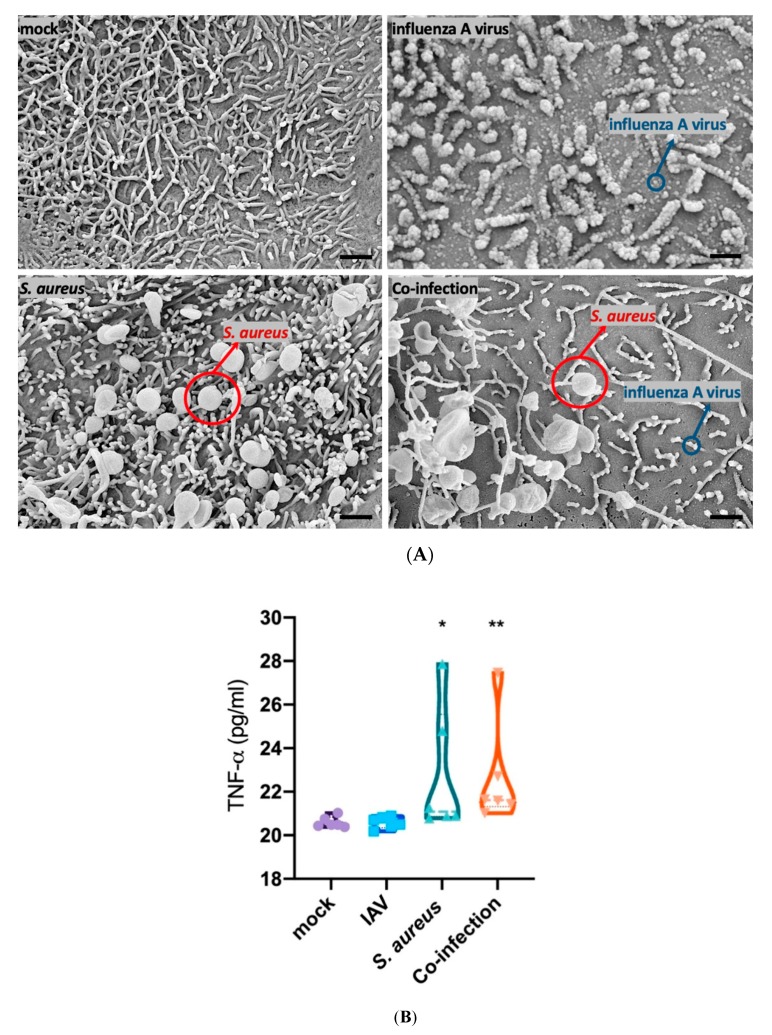
The human alveolus-on-a-chip model was infected with *S. aureus* and /or IAV. (**A**) By using an SEM, it was possible to visualize *S. aureus* (red arrow) and IAV during the budding process (blue arrow), magnification x 10,000, scale bar represents 1 µm. (**B**) TNF-α (pg/mL) is upregulated and released into the supernatant during *S. aureus* mono- and co-infection measured in pg/mL. (**C**) Immunofluorescence staining of the epithelial cell side of the human alveolus-on-a-chip model shows SP-A staining (red) of mock infected, influenza virus, *S. aureus* and co-infected scenarios. Scale bar represents 50 µm. The quantification of the SP-A staining (measured as mean-fluorescence intensity, MFI) reveals decreased intracellular protein levels for *S. aureus* mono- and co-infection (**D**). Mock-treated cells obtained the same medium and treatment as infected cells without the pathogen. Ordinary one-way ANOVA, Tukey´s multiple comparison, * *p* < 0.05 ** *p* < 0.01.

**Figure 5 microorganisms-08-00577-f005:**
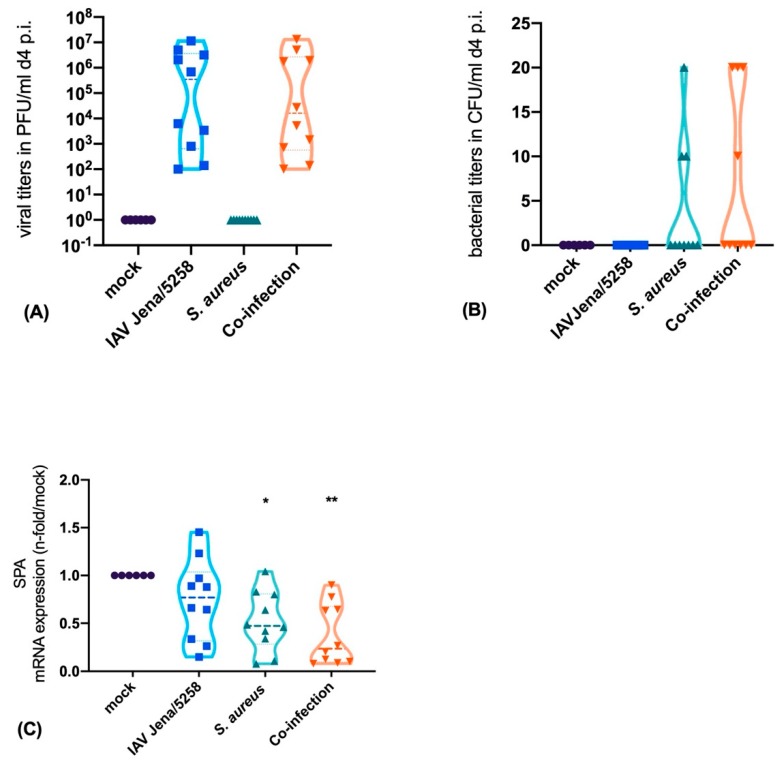
Murine model of *S. aureus* pneumonia showing viral titers (**A**) as well as *S. aureus* colony-forming units (CFUs) (**B**) caused by HA-D222-mpJena/5258, *S. aureus,* and co-infection. Downregulation of SP-A mRNA expression is mainly caused by bacterial mono- and co-infection (**C**). 100x magnification of the lung of C57bl/6 mice, HE-staining, mock-treated without any inflammatory infiltrates, and IAV, *S. aureus* and co-infection of the lung with pro-inflammatory infiltrates (**D**). Mock-treated mice obtained the same medium and treatment as infected mice without the pathogen. Ordinary one-way ANOVA, Tukey´s multiple comparison, * *p* < 0.05 ** *p* < 0.01.

**Table 1 microorganisms-08-00577-t001:** For the qRT-PCR we used gene-specific primers for human GAPDH, TNF-α, SP-A, CD80, CD206, murine GAPDH and SP-A with forward and reverse sequence.

Gene Name	Sequence Forward	Sequence Reverse
***human GAPDH***	5′- CTCTGCTCCTCCTGTTCGAC -3′	5′-CAATACGACCAAATCCGTTGAC -3′
***human TNF-α***	5′- GGAGAAGGGTGACCGACTCA -3′	5′- CTGCCCAGACTCGGCAA -3′
***human SP-A***	5′- GATGGGCAGTGGAATGACAGG -3′	5′- GGGAATGAAGTGGCTAAGGGTG -3′
***RPL37A***	5′- ATTGAAATCAGCCAGCACGC -3′	5′- AGGAACCACAGTGCCAGATCC -3′
***CD80***	5′- TGGTGCTGGCTG GTCTTTC -3′	5′- CGTTGCCACTTCTTTCACTTCC -3′
***CD206***	5′- TCGGGTTTATGGAGCAGGTG -3′	5′- TGAACGGGAATGCACAGGTT -3′
***murine GAPDH***	5′-CAACAGCAACTCCCACTCTTC-3′	5′-GGTCCAGGGTTTCTTACTCCTT-3′
***murine SP-A***	5′-GCAGAGATGGGAGAGATGGTATCAA -3′	5′-ATGGACCTCCATTAGCATGTGGGA-3′

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
