# Peer review of "Staphylococcus aureus Lung Infection Results in Down-Regulation of Surfactant Protein-A Mainly Caused by Pro-Inflammatory Macrophages"

_microorganisms, 2020, doi:10.3390/microorganisms8040577_

Round 1
Reviewer 1 Report
The manuscript by Schicke and colleagues is on an important topic. The authors aimed at describing the regulation of SP-A expression during bacterial (S.aureus) viral (IVA) or co-infection both in vitro and in vivo (mouse model). Unfortunately, the manuscript is quite confusing and misses a lot of important experimental details required to support the obtained results.
Line 48 and 49. A brief description of the innate host immune response in this particular infection should be provided even if it is through the citation of a review. As it is the reader might think that surfactant proteins are the only players involved.
Line 74. Refer if the fetal bovine serum was inactivated (heat inactivated) or not.
Line 85. “Co-infection was performed after 6 days…” The authors started with 6x106 cells but how many are after monocyte differentiation into macrophages. How was the maturation accessed?
Line 90. Check the typo error 5x105
Line 95-96. A brief description should be provided instead of quoting a reference.
Line 98. “The IV strain A Puerto Rico/8 was passed on MDCK.” I assume this is the preparation of viral stocks for infection and should be described.
Line 100. For how long were the bacteria incubated.
Line 101. Explain the meaning of the last sentence. “ The CFU….”
Lines 103-105. The infection time for viruses was 30 minutes but how was done the infection. The virus were added to the cells and incubated or were spin down in a centrifuge. If it was the second the conditions must be described. “…and virus at various multiplicity…” It looks that only two MOI were used. Please rephrase.
Lines 111-113. “After 90 min…” Does this mean that infection time for bacteria were 90 min? Explain why different infections times were used for bacteria and viruses. The cultures were incubated for 20 min with lysosthaphin at each concentration? What was the starting point for the 30min, 4 and 18h? Depending on the starting point the procedures would have been different for at least one time point?
Line 114. TNF-alpha was added for 30 min prior to infection and removed or did it stay until the end of the experiment? Here it looks like that the cells treated with the cytokine were infected. Please explain.
General comment for section 2.3. Explain in detail how co-infection was done and to what corresponds the mock control clearly in each experiment.
Line 122. “…mice were infected with the H1N1….” Explain why in line 98 another terminology was used to describe the virus.
Line 126-127. Describe how the state of health of the animals was monitored.
Line 128. “Secreted cytokines” which ones?
Lines 132-133. The procedure for HE and further microscopic analysis should be described.
Line 135. “… were collected after antibiotic treatment for 30 minutes…” Indicate antibiotic name and concentration. Previously, line 112, the authors mentioned the treatment with lysostaphin. Explain why two different treatments were used. Also explain when the post infection time starts to be counted.
Lines 147-156. It would be easier if the primmers sequences would be in a table instead of the text. The RT-PCR conditions must be described even if briefly.
Line 159. “…previously described.” Insert the section where it is described. If it is not described in the present manuscript introduce it.
Line 168. “The concentration was …” the value of the concentration is missing.
Line 172. “The optical dense….” Did the authors mean density?
Line 173. “Protein amount…” Did the authors mean amount or concentration?
Line 176. “…4h pi…” Only at this time point?
Line 182. Explain when each of the fixation procedures was used.
Line 183-184. The immunolabelling procedure should be described in detail (ab concentrations, incubation times, fluorophore linked to the secondary ab, etc)
Line 185. “…fluorescent mounting….” The fluorescent is a mistake correct?
Line 194. First sentence is not clear please re-arrange to improve clarity.
General comment: The procedure for scanning electron microscopy is missing.
Line 204. “ as well as viable bacteria were recorded…” What did the authors meant by recorded?
General comment to all figure legends including mock. Explain what is this control.
Figure 4. An inset in a higher magnification to highlight S. aureus and IAV would be better. In figure C it would have been advisable to use a counterstaining for the nucleus to allow to compare the total number of cells in each picture.
Figure 5. The images quality and legends are very poor. Better quality images and highlighting of the findings in the image are required.
The authors should considered re-organizing the results section. Discussion and conclusions must be improved.
Reviewer 2 Report
This is a very well written research article on the effects of SP-A expression in macrophages in response to the co-infection of influenza virus and S. aureus. Overall, the article is very well done, the conclusions are supported by the results, the data are clearly presented and there is novelty in the findings included in this research article. I could only point to some minor issues: Line 193, please check the number of this section; Line 194, difficult to read; Figure 5d is not properly cited in the text; the bibliography is quite limited, some more references on the clinical implications of these findings should be added to the reference list and the discussion should expand accordingly to cite those publications.
Round 2
Reviewer 1 Report
The quality of the manuscript was improved nevertheless there are still a few topics that need clarification.
General note: The supplementary data (figures and tables) should be included in Appendixes. Please check the instructions:
Appendix A
The appendix is an optional section that can contain details and data supplemental to the main text. For example, explanations of experimental details that would disrupt the flow of the main text, but nonetheless remain crucial to understanding and reproducing the research shown; figures of replicates for experiments of which representative data is shown in the main text can be added here if brief, or as Supplementary data. Mathematical proofs of results not central to the paper can be added as an appendix
Appendix B
All appendix sections must be cited in the main text. In the appendixes, Figures, Tables, etc. should be labeled starting with ‘A’, e.g., Figure A1, Figure A2, etc.
Lines 82-84. What is relevant for the experiments outcome is whether the serum is or is not inactivated. So please remove the sentence: “The used FBS was not heat-inactivated since the manufacturer already inactivated it.” And in line 82 introduce after 10 % the word inactivated.
Line 92. The answer to the previous question of how many monocytes differentiate into macrophages after 6 days was not answered. The initial number of cells was 6x106 and after 6 days was y cells. The differentiation rate usually is not 100%. In case you observed it please clearly, state that.
Line 164. For the multiplicity of infection (MOI) usually we say how many microorganisms per cell were used. Here a MOI of 5 was used. Please explain if you used 5 bacteria per cell (MOI 5:1) or the other way around. Please apply this through the manuscript (lines 171, 177, etc).
Line 188. The scheme is a nice addition. I have one question: When do you consider your time zero? In other words, is your time zero after the 20 min with lysostaphin?
Lines 200-204. This text is confusing. I assume that TNF-alpha stimulation was used as a “control” or a “comparative” for the expression of m-RNA SPA triggered by pathogen infection. It would be easier to describe this together with “Quantitative real-time PCR” (starting in line 233).
Lines 284 -287: “Afterwards the membranes were incubated overnight at 4°C in PBS containing 0.1% Saponin, 3% donkey serum and anti-SP-A antibodies (Santa Cruz Biotechnologies, Dallas, USA), and fluorescent labeled secondary antibodies (donkey-anti-goat- AlexaFlour647, life technologies, Carlsbad, CA, USA).” The primary and secondary antibodies were added together. Please revise the text.
Did you use methanol and after saponin? Please confirm.
Line 292. In the previous version, the sample was treated with uranyl acetate. This part was removed intentionally or by mistake. If no UA was used please explain if no other contrasting agent was added to the samples.
Figure 4.A Please indicate clearly which structure corresponds to the influenza virus. I would suggest replace the image on the right side upper row by another one in a higher magnification to improve clarity. As it is, is not possible to say that the structure highlighted in this image as IV and the one below are the same.
